# *Escherichia coli* Producing Extended-Spectrum β-lactamases (ESBL) from Domestic Camels in the Canary Islands: A One Health Approach

**DOI:** 10.3390/ani10081295

**Published:** 2020-07-29

**Authors:** Isabel Carvalho, María Teresa Tejedor-Junco, Margarita González-Martín, Juan Alberto Corbera, Vanessa Silva, Gilberto Igrejas, Carmen Torres, Patrícia Poeta

**Affiliations:** 1Microbiology and Antibiotic Resistance Team (MicroART), Department of Veterinary Sciences, University of Trás-os-Montes and Alto Douro, 5000-801 Vila Real, Portugal; isabelbarrosocarvalho@utad.pt (I.C.); vanessasilva@utad.pt (V.S.); 2Department of Genetics and Biotechnology, UTAD, 5000-801 Vila Real, Portugal; gigrejas@utad.pt; 3Functional Genomics and Proteomics Unit, UTAD, 5000-801 Vila Real, Portugal; 4Laboratory Associated for Green Chemistry (LAQV-REQUIMTE), New University of Lisbon, 2829-516 Monte da Caparica, Portugal; 5Area Biochemistry and Molecular Biology, University of La Rioja, 26006 Logroño, Spain; carmen.torres@unirioja.es; 6Research Institute of Biomedical and Health Sciences, University of Las Palmas de Gran Canaria, 35001 Canary Islands, Spain; mariateresa.tejedor@ulpgc.es (M.T.T.-J.); margaritarosa.gonzalez@ulpgc.es (M.G.-M.); juan.corbera@ulpgc.es (J.A.C.); 7Department of Clinical Sciences, University of Las Palmas de Gran Canaria, 35001 Canary Islands, Spain; 8Department of Animal Pathology, Veterinary School, University of Las Palmas de Gran Canaria, 35001 Canary Islands, Spain

**Keywords:** antimicrobial resistance, extended-spectrum β-lactamases, *Escherichia coli*, camels

## Abstract

**Simple Summary:**

Extended-spectrum beta-lactamase (ESBL) producing *Escherichia coli* is an important problem in hospital settings. Camels are known to harbor multidrug-resistant Gram-negative bacteria and to be involved in the transmission of various microorganisms to humans. Fecal samples of 58 camels were recovered in the Canary Islands for detection and characterization of cefotaxime-resistant (CTX^R^) and ESBL-producing *E. coli* isolates. Five samples carried CTX^R^
*E. coli* isolates and two of them contained ESBL-positive *E. coli* (3.4%) with the following characteristics: (ESBL/phylogroup/sequence type): CTX-M-15/A/ST3018 and CTX-M-15/B1/ST69. The three remaining isolates recovered from CTX-supplemented plates were ascribed to phylogroup-B_1_. Due to the participation of these animals in touristic activities in the region, the potential transference of ESBL-positive bacteria between humans and animals could happen and should be further monitored.

**Abstract:**

Objective: This work aimed to determine the carriage rate of ESBL-producing *Escherichia coli* as well as their genetic characteristics in camels from the Canary Islands, Spain. Methods: Fecal samples were recovered from 58 healthy camels from Gran Canaria (n = 32) and Fuerteventura Islands (*n* = 26) during July 2019. They were seeded on MacConkey (MC) agar no supplemented and supplemented (MC + CTX) with cefotaxime (2 µg/mL). Antimicrobial susceptibility was determined by disk diffusion test (CLSI, 2018). The presence of *bla*_CTX-M_, *bla*_SHV_, *bla*_TEM,_
*bla*_CMY-2_ and *bla*_OXA-1/48_ genes was tested by PCR/sequencing. Furthermore, the *mcr-*1 (colistin resistance), *tet*A/*tet*B (tetracycline resistance), *int*1 (integrase of class 1 integrons) and *stx*_1,2_ genes were analyzed. Phylogenetic groups and sequence types were determined by specific-PCR/sequencing for selected isolates. Results: *E. coli* was obtained from all the 58 camels in MC media (100%) and in five of them in MC + CTX media (8.6%). Furthermore, 63.8% of *E. coli* isolates recovered from MC agar were susceptible to all the antibiotics tested. The five *E. coli* isolates recovered from MC + CTX media were characterized and two of them were ESBL-producers (3.4%). Both ESBL-producer isolates carried the *bla*_CTX-M-15_ gene and belonged to the lineages ST3018 (phylogroup A) and ST69 (phylogroup B1). The 3 ESBL-negative isolates recovered from MC-CTX plates were ascribed to phylogroup-B_1_. Conclusions: Camels can be a source of ESBL-producer bacteria, containing the widespread *bla*_CTX-M-15_ gene associated with the lineages ST3018 and ST69.

## 1. Introduction

During the last decade, different studies have been published in Europe regarding antimicrobial resistance among Enterobacteriaceae in wild animals [1,2,3,4], pets [5,6,7,8,9] and humans [10,11,12,13]. The excessive and sometimes inappropriate antibiotic use in both human and veterinary medicine has been considered as one of the main contributors of the selection and dissemination of multidrug-resistant (MDR) bacteria [14,15]. According to the World Health Organization (WHO) [16], the abuse and misuse of these drugs can also lead to this global health concern and compromise prevention and treatment of an increasing range of infections caused by bacteria.

*Escherichia coli,* belonging to the Enterobacteriaceae family, is a normal inhabitant of the human intestine, but at the same time, it is an important opportunistic pathogen associated with severe sepsis and urinary infections, among other infections at hospital level [17,18]. In the case of camels, diarrhea and other infectious diseases are considered to be the main causes of economic loss associated with poor growth, medication costs, and animal death [19]. In the last few years, the number of researches on the epidemiology of MDR bacteria has increased in both hospital and community settings around the world, with special attention to extended-spectrum β-lactamase (ESBL), plasmid mediated AmpC β-lactamase (pAmpC), and carbapenemase production in Enterobacteriaceae [20]. The genes encoding these enzymes are frequently plasmid-located and can be horizontally transferred to other bacteria [21]. The spread of *E. coli* isolates producing CTX-M-type beta-lactamases is mostly responsible for the increased incidence of ESBL, especially CTX-M-15 variant, both in animals and humans [18,22,23,24,25].

Regarding the collaborative and multi-disciplinary “One Health” approach, the interaction between animal, human, and environmental health is required. In this line, Canarian camels are in contact with humans, and both with the environment, disseminating the presence of potential resistant bacteria between the three axes of One Health.

To our knowledge, this is the first report on ESBL-producing bacteria performed in camels from Europe, specifically, the Canary Islands. Even though antibiotic resistance in camels has been less studied than in other domestic animals worldwide [26,27,28,29,30], they might have an important role in the spread of ESBL/pAmpC bacteria [19,24,31,32]. Due to the participation of camels in touristic activities in the region, the close interaction between camels and humans facilitates the potential transference of ESBL-positive bacteria, highlighting the need for further work in this area. The aim of this research was to determine the carriage rate of ESBL-producing *E. coli* as well as their genetic lineages in camels from Fuerteventura and Gran Canaria (Canary Islands), Spain.

## 2. Materials and Methods

### 2.1. Animals and Sampling

A total of 58 fecal samples were recovered from apparently healthy camels (*Camelus dromedaries*) used in touristic activities from Gran Canaria (*n* = 32) and Fuerteventura (*n* = 26) Islands during July 2019 (Appendix A). It is important to note that these animals were in contact with humans in touristic activities (domestic animals) and the camels from the same island were in contact with each other. One fecal sample per animal was obtained rectally using a sterile cotton swab during a veterinarian intervention and all samples were dispatched immediately to the Research Institute of Biomedical and Health Sciences (University of *Las Palmas de Gran Canaria*, Spain).

### 2.2. Escherichia coli Isolation

The fecal samples were inoculated onto MacConkey agar plates non supplemented (MC) and supplemented with 2 µg/L of cefotaxime (MC + CTX) for cefotaxime resistant (CTX^R^) *E. coli* recovery. The plates were incubated for 24 h at 37 °C and colonies growing with a typical morphology for *E. coli* (red or pink, non-mucoid colonies) were recovered; they were identified by classical biochemical methods named IMViC (Indol, Methyl-red, Voges–Proskauer and Citrate) and also for Analytical Profile Index (API 20E gallery). In order to confirm the bacterial species identification, the matrix-assisted laser desorption/ionization time-of-flight mass spectrometry method (MALDI-TOF MS, Bruker) was applied in the Laboratory of Biochemistry and Molecular Biology in the University of La Rioja (Logroño, Spain). One *E. coli* per sample on each of the media used was kept and characterized by genetic methods.

### 2.3. Susceptibility Testing

Antimicrobial susceptibility testing was performed on Mueller-Hinton agar using the disk diffusion method, according with Clinical Laboratory Standards Institute standard guidelines CLSI, 2018^33^. *E. coli* isolates were tested against the following antimicrobial agents (μg/disk): ampicillin (10), amoxicillin + clavulanic acid (20 + 10), cefoxitin (30), cefotaxime (30), ceftazidime (30), aztreonam (30), imipenem (10), tetracycline (30), ciprofloxacin (5), trimethoprim-sulfamethoxazole (1.25 + 23.75), gentamicin (10), tobramycin (10), streptomycin (10), and amikacin (30). The strains were recorded as susceptible, intermediate, or resistant according to the zone diameter interpretative standards recommended by CLSI (2018) [33]. The detection of ESBL production was carried out using three disks of antibiotics in the same line: cefotaxime, ceftazidime, and amoxicillin/clavulanic acid (CLSI, 2018)^33^. Based on the disc diffusion method for these three antimicrobials, we concluded if the isolate were ESBL positive when was visible the ghost zone between these antibiotics.

### 2.4. DNA Extraction and Quantification

Genomic DNA from MDR strains was extracted using the InstaGene Matrix (BioRad, Hercules, CA, USA), according to the manufacturer’s instructions. In order to quantify the DNA concentration and the level of purity, the absorbance readings were taken at 260 and 280 nm (Spectrophotometer ND-100, Nanodrop Thermo Fisher Scientific, Wilmington, DE, USA).

### 2.5. Antibiotic Resistance Genes

The genetic basis of resistance was investigated using PCR/sequencing of the obtained amplicons. Positive controls of the University of La Rioja (Logroño, Spain) were used in this study. 

The presence of the *bla*_CTX-M_ (groups 1 and 9), *bla*_SHV_, *bla*_TEM_, *bla*_OXA-1,_
*bla*_OXA48,_
*bla*_CMY-2_, and *bla*_KPC_ genes [34] was tested by PCR/sequencing [35,36]. It is important to note that CTX-M genes were tested for ESBL-positive isolates while the isolates with the typical AmpC phenotype, the presence of *bla*_CMY-2_ gene was tested. Furthermore, the *mcr-*1 (colistin resistance), *tet*A/*tet*B (tetracycline resistance), *int*1 genes (integrase of class 1 integrons), as well as the *stx*_1,2_ genes were also analyzed. Phylogenetic classification of *E. coli* isolates was performed according to the existence of *chu*A, *yja*A, and TSPE4.C2 [37].

### 2.6. Multilocus Sequence Typing of E. coli Isolates 

The multilocus sequence typing (MLST) with seven housekeeping genes (*icd*, *fum*C, *mdh*, *adk*, *rec*A, *pur*A, and *gyr*B) was carried out in the two ESBL-producing isolates, according to the protocol for *E. coli* on the PubMLST site [38]. The allele combination was determined after sequencing the seven genes, and the sequence type (ST) and clonal complex (CC) was identified.

## 3. Results and Discussion

### 3.1. Characteristics of E. coli Isolates Obtained in Non-Supplemented Media (MC)

*E. coli* isolates were recovered from all fecal samples in MC plates, and a collection of 58 isolates were obtained (one/sample). The percentages of antimicrobial resistance in these *E. coli* isolates are presented in Table 1. Most of our isolates were susceptible to all antibiotics tested (*n* = 37, 63.8%) and only two isolates (5.2%) showed a MDR phenotype (resistance to at least three families of antibiotics). Resistance to ampicillin was detected in 34.5% of isolates and to streptomycin in 10.3%. None of the isolates obtained in MC plates (without antibiotic supplementation) showed resistance to amoxicillin-clavulanic acid, broad spectrum cephalosporins (cefotaxime and ceftazidime) aminoglycosides (except streptomycin), or imipenem. Moreover, all these isolates showed susceptibility to ciprofloxacin (Table 1).

To our knowledge, there were few reports in which fecal *E. coli* isolates, recovered without antibiotic selection from healthy camels, were studied for antimicrobial resistance. In this line, *E. coli* isolates were detected in fecal samples from apparently healthy camels in Bangladesh [39] and, in contrast, from diarrheic camel calves in Saudi Arabia [40]; however, they lacked data about antibiotic resistance genes.

### 3.2. Characteristics of E. coli Obtained in Supplemented Media (MC-CTX)

*Escherichia coli* isolates were recovered in five of the 58 tested samples, when they were cultured in MC + CTX plates. Table 2 shows the characteristics of these isolates.

All of these five isolates were resistant to ampicillin, cefotaxime, and ceftazidime and three of them also to amoxicillin + clavulanic acid. Resistance to tetracycline and trimethoprim + sulfamethoxazole was found in one and two isolates, respectively (Table 2). We also detected one strain with intermediate resistance to imipenem (isolate X2263), although it lacked the *bla*_OXA48_ and *bla*_KPC_ genes.

Two of the CTX^R^
*E. coli* isolates were ESBL-producers, representing 3.4% of the total animals tested. A higher prevalence of ESBL-positive *E. coli* (26.9%) was found in a previous study in camels from Saudi Arabia [41]. One of the ESBL-producing *E. coli* isolates of our study was obtained from a camel of Gran Canaria and it showed the following resistant genotype: *bla*_CTX-M-15_, *bla*_TEM-1_, *tet*A, and *tet*B; this isolate also carried the *int* gene encoding the integrase of class 1 integrons. Moreover, the other ESBL-positive isolate was collected from a camel of Fuerteventura and it was positive for *bla*_CTX-M-15_ gene (Table 2). The detection of ESBL-producing isolates in camels in this study could represent a public health concern if transmitted to humans [19]. These results are similar to the data obtained in the same animal species in a previous study, in which three ESBL-*E. coli* isolates carried *bla*_CTX-M-15_ (*n* = 2) and *bla*_CTX-M-1_ genes (*n* = 1) [24]. Furthermore, Ben Sallem, et al. [42] detected the presence of *bla*_CTX-M-1_ in one isolate from a dromedary in Tunisia.

The three CTX^R^ and ESBL-negative isolates recovered in MC + CTX plates of our study belonged to the B_1_ phylogroup, and they did not carry any of the beta-lactamase genes tested, including *bla*_CMY-2_ (Table 2); these strains might carry other non-tested acquired AmpC beta-lactamases, or could present hyperproduction of the chromosomal AmpC beta-lactamase [43].

The *mcr*-1 determinant, encoding colistin resistance, was studied in all *E. coli* isolates of this work, and all of them were *mcr-1* negative. However, other studies done in camels from different African cities found *E. coli* harboring the *mcr*-1 gene [24,32]. None of our isolates were positive for the *stx*1 or *stx*2 genes, contrasting with the results obtained in a recent study done on fecal samples of African dromedary camels [44].

The two CTX-M-15-positive isolates were ascribed to the sequence type ST3018 (phylogroup A) and ST69 (phylogroup B1; Table 2). These results agree with the ones of Saidani, et al. [24] and Bessalah, et al. [19], suggesting the predominance of commensal *E. coli* isolates (phylogroups A and B_1_) in healthy camels from Tunisia. Furthermore, these results are in line with the detection of B_1_ phylogroup in the *E. coli* isolated from fecal sample from a dromedary in Tunisia [42].

This unusual ST lineage (ST3018) had not been reported before in either humans or in camels’ isolates. There are only two recent reports on ST3018 *E. coli*, one on isolates from dairy cattle in the USA [45] and another one on isolates from a poultry farm in Ghana [46], associated with the *bla*_CTX-M-15_ gene. Additionally, our data revealed the presence of ST3018-CTX-M-15 *E. coli*, which should be considered a recent lineage detected in different ecosystems with relevance for animals and with potential transmission to humans (Table 2).

Regarding the lineage ST69, detected in a CTX-M-15-positive *E. coli* isolate in this study, it has been previously referred to as a pandemic clonal lineage in humans implicated in extraintestinal infections [45]. The ST69 clone was previously reported as well among dairy cattle-associated *E. coli* carrying *bla*_CTX-M-65_ gene in the Washington State [45]. Our results are also in line with the detection of ST69/*bla*_CTX-M-1,15_ among dogs and cattle in Europe [47]. Furthermore, a recent study showed the presence of ST69/ *bla*_CTX-M-14,15_ among febrile urinary tract infections in children from France [48]. Interestingly, the dogs and humans of the family carried an identical CTX-M-group-9 *E. coli* ST69 strain, indicating interspecies transmission [49]. The same lineage was also present in both avian (gulls) and human populations in Chile [50].

Very few reports have been produced on camels concerning the prevalence and characterization of ESBL-producer *E. coli* in the fecal microbiota of healthy camels, and most of the studies have focused on *E. coli* isolates in other domestic animals [51,52,53] or in camels only from the Africa continent [19,24,41].

To our knowledge, this is the first study related to commensal ESBL-producing *E. coli* isolates recovered from camels in Spain (Canary Islands) and also the first report in Europe. This research is also an important contribution as a One Health approach.

## 4. Conclusions

This study demonstrated that camels could become a source of ESBL-producing *E. coli* isolates associated with recent clones ST3018 and ST69. For the first time in Spain and Europe, our study reported the presence and possible dissemination of the *bla*_CTX-M-15_ gene in *E. coli* isolates of fecal samples of camels in the Canary Islands, which could potentially be transferred to close-contact humans. 

These resistant bacteria should be monitored in the future, mostly in this type of animals that lives in such close contact with humans. Further research with a larger number of camels will be necessary to elucidate the role of *E. coli* in healthy and sick camels.

## Figures and Tables

**Table 1 animals-10-01295-t001:** Resistance phenotype of *E. coli* isolates recovered of fecal samples of healthy camels from MC plates (non-supplemented with cefotaxime).

Resistance Phenotype ^a^	Number of Isolates	Origin (Number of Samples) ^b^
Susceptible	37	GC (15), FV (22)
AMP	15	FV (3), GC (12)
AMP, S	3	GC (3)
S	1	GC
AMP, S, TET, SXT	1	GC
AMP, S, TET	1	FV

Legend: ^a^ AMP—ampicillin; S—streptomycin; TET—tetracycline; SXT—trimethoprim-sulfamethoxazole; ^b^ GC—Gran Canaria; FV—Fuerteventura.

**Table 2 animals-10-01295-t002:** Antimicrobial resistance phenotype and genotype of *Escherichia coli* isolates recovered of fecal samples of healthy camels from MC + CTX plates (supplemented with cefotaxime).

Isolate Number	Origin ^a^	ESBL ^b^	Resistance Phenotype ^c^	β-Lactamases	Resistance Genes	PG ^d^	MLST ^e^
**X1848**	GC	P	AMP, CTX, CAZ, TET, S, SXT	CTX-M-15, TEM-1	*tet*A, *tet*B, *int*1	A	ST3018
**X2263**	FV	P	AMP, AMC, CTX, CAZ, SXT	CTX-M-15	-	B1	ST69
**X1929**	GC	N	AMP, FOX, CTX, CAZ	-	-	B1	
**X1849**	FV	N	AMP, AMC, FOX, CTX, CAZ	-	-	B1	
**X1850**	FV	N	AMP, AMC, CTX, CAZ	-	-	B1	

Legend: ^a^ GC—Gran Canaria; FV—Fuerteventura; ^b^ P—Positive, N—Negative; ^c^ AMP—ampicillin; CTX—cefotaxime; CAZ—ceftazidime; TET—tetracycline; S—streptomycin; *STX*—trimethoprim-sulfamethoxazole; AMC—amoxicillin–clavulanic acid; FOX—cefoxitin. ^d^ PG—Phylogroup; ^e^ MLST—MultiLocus Sequence Typing.

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
