# Peer review of "Escherichia coli Producing Extended-Spectrum β-lactamases (ESBL) from Domestic Camels in the Canary Islands: A One Health Approach"

_animals, 2020, doi:10.3390/ani10081295_

Round 1

Reviewer 1 Report

This is a good characterization of ESBL-producing E. coli  in camels from Canary Islands, Spain. This is an important addition to the current knowledge and it has been written with excellent interpretation and understanding.

Author Response

This is a good characterization of ESBL-producing E. coli in camels from Canary Islands, Spain. This is an important addition to the current knowledge and it has been written with excellent interpretation and understanding.

Answer: Thank you so much for the feedback.

Reviewer 2 Report

Dear Authors,

In my opinion your manuscript covers a very interesting topic – antimicrobial resistance in wildlife and the possible transmission of resistant genes from wildlife to humans, tackling the importance of the One Health approach. The manuscript is overall well written, but some sentences are not clear and need clarification. Despite the appropriateness of the idea, at its essence, I think that the authors extrapolated the scope of the MS; starting with the title that is far too ambitious. I think that due to the importance of your findings, there is obvious a clear link with the “One Health” concept. However, this is indirect as the authors are not exploring the links and routes of transmission of AMR in the 3 axis of One Health. So I would suggest the authors to focus the introduction on the real value of the paper. Also, as it stands, I don’t think that the ideas are developed enough for a full article. I would rather see this as a short report.  I guess that the main ideas are there; it is just a question of having a clear line to follow. Also, you should give more emphasizes on clear-cut points, in both the introduction and the discussion (WHY is this or that important?).So, I suggest the authors to spend some time in developing the manuscript.

Although I found the work interesting I believe that the article must be improved to be up to its goals.

I hope these comments will help.

Author Response

Dear Authors,

In my opinion your manuscript covers a very interesting topic – antimicrobial resistance in wildlife and the possible transmission of resistant genes from wildlife to humans, tackling the importance of the One Health approach. The manuscript is overall well written, but some sentences are not clear and need clarification. Despite the appropriateness of the idea, at its essence, I think that the authors extrapolated the scope of the MS; starting with the title that is far too ambitious. I think that due to the importance of your findings, there is obvious a clear link with the “One Health” concept. However, this is indirect as the authors are not exploring the links and routes of transmission of AMR in the 3 axis of One Health. So I would suggest the authors to focus the introduction on the real value of the paper. Also, as it stands, I don’t think that the ideas are developed enough for a full article. I would rather see this as a short report.  I guess that the main ideas are there; it is just a question of having a clear line to follow. Also, you should give more emphasizes on clear-cut points, in both the introduction and the discussion (WHY is this or that important?). So, I suggest the authors to spend some time in developing the manuscript. Although I found the work interesting I believe that the article must be improved to be up to its goals. I hope these comments will help.

Answer: Thank you for the comments. We change the title of the manuscript to be related with a One Health approach “Escherichia coli producing Extended-Spectrum β-lactamases (ESBL) from domestic camels in Canary Islands: A One Health approach. “

More information was also added in the introduction (lines 79-82) and discussion (lines 235-237) sections. It is correct that we have small number of samples analyzed in this study. Despite of it, and regarding the type of the report, we considered that this article is more indicated to be a research report, according with its new in the field: Escherichia coli producing Extended-Spectrum β-lactamases (ESBL) from camels.

Reviewer 3 Report

The authors first reported that Extended-Spectrum β-lactamases (ESBL)-positive E. coli from camels in Canary Islands, Spain. They obtained 58 E. coli isolates recoverd from the faecal samples of 58 apparently healthy camels in MC media (one/sample) and in 5 of them in MC+CTX media. 63.8% of E. coli isolates recovered from MC agar were sensible to all the antibiotics tested. The five E. coli isolates recovered from MC+CTX media were characterized and 2 of them were ESBL-producers (3.4%). Both ESBL-producer isolates carried the blaCTX-M-15 gene and belonged to the lineages ST3018 (phylogroup A) and ST69 (phylogroup B1). The 3 ESBL-negative isolates recovered from MC-CTX plates were ascribed to phylogroup-B1.They finally concluded that the dissemination of the blaCTX-M-15 gene in E. coli isolates of fecal samples of camels in Canary Islands, which could potentially be transferred to close-contact humans. The manuscript is well written and only minor revision would be required.Detailed comments:1. Line 142: “genotype” should be deleted.2. Line 148: The format is different with that of others subhead.3. Line 164: “sulfamethoxazole was” instead of “sulfamethoxazole-was”.4. Line 169: “ESBL-producing E. coli” instead of “ESBL producing-E. coli”.

  1. All the references should be written in one format.

Author Response

The authors first reported that Extended-Spectrum β-lactamases (ESBL)-positive E. coli from camels in Canary Islands, Spain. They obtained 58 E. coli isolates recovered from the faecal samples of 58 apparently healthy camels in MC media (one/sample) and in 5 of them in MC+CTX media. 63.8% of E. coli isolates recovered from MC agar were sensible to all the antibiotics tested. The five E. coli isolates recovered from MC+CTX media were characterized and 2 of them were ESBL-producers (3.4%). Both ESBL-producer isolates carried the blaCTX-M-15 gene and belonged to the lineages ST3018 (phylogroup A) and ST69 (phylogroup B1). The 3 ESBL-negative isolates recovered from MC-CTX plates were ascribed to phylogroup-B1.They finally concluded that the dissemination of the blaCTX-M-15 gene in E. coli isolates of fecal samples of camels in Canary Islands, which could potentially be transferred to close-contact humans. The manuscript is well written and only minor revision would be required.

Detailed comments:1. Line 142: “genotype” should be deleted.2. Line 148: The format is different with that of others subhead.3. Line 164: “sulfamethoxazole was” instead of “sulfamethoxazole-was”.4. Line 169: “ESBL-producing E. coli” instead of “ESBL producing-E. coli”.

All the references should be written in one format.

Answer: The authors acknowledge the complement to our article. Detailed minor revisions were done directly in the manuscript:

  1. Line 142: “genotype” should be deleted.

The change was done in the manuscript.

  1. Line 148: The format is different with that of others subhead.

We adjust the text margins related to table 1 title and we change the letter type (Times New Roman to Palatino) in the table 2.

  1. Line 164: “sulfamethoxazole was” instead of “sulfamethoxazole-was”.

The change was done in the manuscript.

  1. Line 169: “ESBL-producing E. coli” instead of “ESBL producing-E. coli”.

The change was done in the manuscript.

Reviewer 4 Report

The aim of the study was to assess the occurrence and molecular characterization of E. coli strains isolated from camels

It should be stated that the concept of the study is correct and desirable due to the potential close contact of this group of animals with humans, but I would like to address a few comments:

- the title is somewhat misleading: please specify what the 'first European report' refers to: whether the group of animals studied (only this report is talking about camel testing in Europe in terms of ESBL E. coli producers?). Camels are not very popular as breeding animals in this part of the world, therefore the title is not so clear

Line 47: please replace by „susceptible”

Line 83-88 The procedure requires a more detailed description. It may not be obvious to all readers what camel species are found on the Canary Islands; in scientific studies the Latin name of the species should be given. As I understand, a camel is considered as  domestic animal, so the ethics committee's approval was not required, but it would be worthwhile to indicate under what conditions the samples were taken (e.g. during a clinical examination by a veterinarian?). Was there a key according to which the animals were selected (they had contact with each other, or were they from different places on the islands?)

Line 98: Were strains randomly selected from each positive-culture plate?

Line 102: Standard is not synonymous with guideline. If the test was performed exactly as presented in the CLSI standard, please specify the nomenclature; if no, please indicate changes in the procedure.

Line 109: please specify whether tests for Extended-Spectrum β-Lactamases were carried out according to Table 3A (CLSI, 2018) or was it based on the results of disc diffusion methods for these three antimicrobials?

Line 121: Why the authors did not use the newer and more precise method of phylotyping? (The Clermont Escherichia Coli Phylo-Typing Method Revisited: Improvement of Specificity and Detection of New Phylo-Groups)

Line 136: It is not very clear why the authors mentioned insuseptibilty only to ciprofloxacin since the tested strains were also susceptible to aminoglycosides except streptomycin - please clarify this sentence.

Lines 137-141. Despite the lack of data on resistance genes in the studies of other authors, it would be worthwhile to compare at least phenotypic resistance with the results of studies of other authors in the discussion. Discussion of the results of E.coli from non-supplemented media is poor.

Line 149: Please replace "seeded" with "cultured"

Line 179: „Materials and methods” section should be consistent with the discussion of the results. In the previous section, the authors did not describe that they tested the blaCMY-2 gene. It is also not entirely clear how phenotypically differentiated the classic ESBL and AmpC type (please see comment to line 109). Please clearly define this part of the experiment

Lines 189-190: Too far-reaching statement: the authors defined the fylogroup for only 5 strains and this is not a very representative group to suggest predominace of A and B1 - I suggest rewording

Line 196: Three strains of ST 3018 blaCTXM 15 (ncluding one from this study) does not quite match the statement "widely distributed"

Line 210-211 Finally, the authors genetically analyzed only 5 strains (including MLST for only two), so I suggest to slightly soften the term "Genetic background in commensal E. coli ........... ..".

I agree that this is the first report in Europe but it must be taken into account that only in a few European countries camels are considered in the category of domestic animals, rather they are mostly in category at least exotic animals, so I suggest changing the manuscript title and meaning of this sentence

Line 215: "dissemination" or only "presence" at this stage of studies and results?

Author Response

The aim of the study was to assess the occurrence and molecular characterization of E. coli strains isolated from camels

It should be stated that the concept of the study is correct and desirable due to the potential close contact of this group of animals with humans, but I would like to address a few comments:

Answer: Thank you for the useful comments. We did the changes in the manuscript, as following:

  • the title is somewhat misleading: please specify what the 'first European report' refers to: whether the group of animals studied (only this report is talking about camel testing in Europe in terms of ESBL E. coli producers?). Camels are not very popular as breeding animals in this part of the world, therefore the title is not so clear

Answer This is the first study done in camels from Europe, and from Canary Islands, in which ESBL producing was detected. It was clarified in the manuscript.

  • Line 47: please replace by „susceptible”

Answer:  The change was done in the manuscript.

  • Line 83-88 The procedure requires a more detailed description. It may not be obvious to all readers what camel species are found on the Canary Islands; in scientific studies the Latin name of the species should be given. As I understand, a camel is considered as domestic animal, so the ethics committee's approval was not required, but it would be worthwhile to indicate under what conditions the samples were taken (e.g. during a clinical examination by a veterinarian?). Was there a key according to which the animals were selected (they had contact with each other, or were they from different places on the islands?)

Answer:  Thank you for the useful comment. The information was added in the “Animals and sampling” section of the manuscript (lines 97-102).

Every year the Canary Islands’ animal health authorities are obliged to carry out a livestock operation to combat tuberculosis among other diseases. This EU official directive is applied also to dromedaries in the Canary Islands. Then, all the animals have to pass through the handling system at the moment of reading of the intradermal tuberculin test completed by an official veterinarian. One member of our team (Dr. Corbera, who is also a Vet) carried out the sampling at the same moment of the official operation, in a random manner in the two farms included in this study. Both owners signed an agreement with the ULPGC and an authorization for the sampling in their animals.

  • Line 98: Were strains randomly selected from each positive-culture plate?

Answer:  The faecal samples from camels were seeded on MacConkey agar. The colonies with the typical morphology of E. coli were selected randomly and their identification was confirmed by Maldi-TOF. Furthermore, their DNA was extracted and analyzed by genetics methods.

  • Line 102: Standard is not synonymous with guideline. If the test was performed exactly as presented in the CLSI standard, please specify the nomenclature; if no, please indicate changes in the procedure.

Answer: We would like to say that we follow the CLSI standard guidelines to classify the level of antibiotic resistance in each isolate. The change was done directly in the manuscript.

  • Line 109: please specify whether tests for Extended-Spectrum β-Lactamases were carried out according to Table 3A (CLSI, 2018) or was it based on the results of disc diffusion methods for these three antimicrobials?

Answer: We seeded the bacteria in Mueller Hinton agar and put the three antibiotics in the same line. Twenty-four hours after, we read the results. When the ghost zone was visible, we concluded that this isolate is ESBL-positive. The diameter of each antibiotic was measured for each sample. The information was clarified in lines 125 and 126 of the manuscript.

  • Line 121: Why the authors did not use the newer and more precise method of phylotyping? (The Clermont Escherichia Coli Phylo-Typing Method Revisited: Improvement of Specificity and Detection of New Phylo-Groups)

Answer: We used the Clermont standard guidelines (2000) based on the presence/absence of the three genes chuA, tsp and yja (groups A, B1, B2 and D), detected by PCR. This method is standardized in our laboratory, and in this line, other comparative studies can be carried out with strains from different origins studied by the same authors.

  • Line 136: It is not very clear why the authors mentioned insuseptibilty only to ciprofloxacin since the tested strains were also susceptible to aminoglycosides except streptomycin - please clarify this sentence.

Answer: Thank you for the advice. The information was changed and clarified in the manuscript.

  • Lines 137-141. Despite the lack of data on resistance genes in the studies of other authors, it would be worthwhile to compare at least phenotypic resistance with the results of studies of other authors in the discussion. Discussion of the results of coli from non-supplemented media is poor.

Answer: As we refer in the text, there are lack of data related to antibiotic resistance in camels from Europe. We don’t have previous literature to compare directly this information, so we tried to compare with data from camels in Africa. Regarding the specific studies of Adamu 2018 and Baschura 2019, only the sxt genes were analyzed; in the same line, Rhouma 2018 only studied the mcr gene. However, we add more information in the discussion section of the manuscript, directly associated with the studies done by Bessalah 2016 and Saidani 2019 (just in MacConkey media supplemented with antibiotic), both with camels from Tunisia.

  • Line 149: Please replace "seeded" with "cultured"

Answer: The change was done in the text.

  • Line 179: „Materials and methods” section should be consistent with the discussion of the results. In the previous section, the authors did not describe that they tested the blaCMY-2 gene. It is also not entirely clear how phenotypically differentiated the classic ESBL and AmpC type (please see comment to line 109). Please clearly define this part of the experiment

Answer: The blaCMY-2 gene was added in material and methods section (line 134) and also in the abstract. We tested first the CTX-M-universal and CTX-M-group 1 genes for ESBL positive isolates; the blaCMY-2 gene associated with AmpC gene was tested in ESBL negative isolates with CTX resistance. This information was added in the manuscript (lines 135-137).

Phenotypically, the isolates with ESBL or acquired AmpC beta-lactamases are clearly differentiated. In the case of ESBL producers, they present the augmentation in the inhibition halo of cefotaxime and ceftazidime in the proximity of amoxicillin-clavulanic acid. In this case, isolates maintain susceptibility to ceftazidime. On the other hand, the isolates with acquired AmpC beta-lactamase do not show the ESBL phenotypic character but they are AMC and ceftazidime resistant. Regarding the isolates with the typical AmpC phenotype, the presence of blaCMY-2 gene was tested.

  • Lines 189-190: Too far-reaching statement: the authors defined the fylogroup for only 5 strains and this is not a very representative group to suggest predominance of A and B1 - I suggest rewording

Answer: The aim of this study is analyzing only the isolates showing cefotaxime-resistance which grow in MacConkey agar supplemented with this antibiotic. It is true that 5 strains are not representative, but these are the isolates that we are interested in analyzing.

  • Line 196: Three strains of ST 3018 blaCTXM 15 (ncluding one from this study) does not quite match the statement "widely distributed"

Answer: Thank you for the comment. We refer that this lineage is a genetic line detected in different ecosystems because it was found in dairy cattle in USA (reference 45), in poultry farm in Ghana (reference 46) and in our study in Canary Islands. The information was clarified in the manuscript.

  • Line 210-211 Finally, the authors genetically analyzed only 5 strains (including MLST for only two), so I suggest to slightly soften the term "Genetic background in commensal E. coli ........... ..".

Answer: We really appreciated your comment. We changed the information in the text (lines 235 and 236).

  • I agree that this is the first report in Europe but it must be taken into account that only in a few European countries camels are considered in the category of domestic animals, rather they are mostly in category at least exotic animals, so I suggest changing the manuscript title and meaning of this sentence

Answer: There are few studies related to domestic animals in Europe (only in Canary Islands). For this reason, we considered to change the title and the suggestion was added.

  • Line 215: "dissemination" or only "presence" at this stage of studies and results?

Answer: The information was clarified in the text as “presence and possible dissemination”.

Reviewer 5 Report

The paper title “First European Report of Escherichia coli producing  Extended-Spectrum β-lactamases (ESBL) from camels in Canary Islands: Animal-Human interface in antibiotic resistance” is an interesting article that provides new and important insights about ESBL spread isolated from camels on the Canary Islands, but the article needs a big, careful revision from the Authors. A lot of valuable work was done but the article has a lot of misunderstandings and mistakes. General comments are that the Authors should read the Instructions for Authors in “Animals” and prepare an article according to them eg. e-mail after affiliation, in “Abstract”, is a reference in line 41, the References are wrong cited in the text: should be [1-4], line 57.

“Material and Methods”

Animals and sampling” – “One faecal sample per animal was obtained rectally” what with Ethical proof ?

“Escherichia coli isolation” I understand that the Authors put faecal samples from swabs on MacConkey ? I need a few words of explanation of this procedure.

“Antibiotic resistance genes” ” as well as the sxt1,2 genes were also analyzed” this “sxt1,2” I once again need a few words of explanation

All article needs English correction. For detailed comments, see minor comments 

Minor comments

Line 4; Canary Islands” the authors should check “the Canary Islands”

Line 26; abbreviation Escherichia coli (E. coli)

Line 27; the spelling of “harbor” is a non-British variant, the authors should check if in all text should be “harbour”

Line 27; “multidrug resistant” is missing a hyphen

Line 29; “Canary Islands” the authors should check “the Canary Islands”

Line 36; “Objective: The aim of this work was” “This work aimed”

Line 37; “Canary Islands” the authors should check “the Canary Islands”

Line 38; the authors should remove “apparently” it is not necessary in this place

Line 41; The authors should check “the disk diffusion”

Line 43; What the authors meant by “sxt1,2

Line 45; “Escherichia coli” The authors should decide abbreviation or not Line 36-37

Line 52; proposition “associated to” should be “associated with”

Line 59; “multidrug resistant” is missing a hyphen “multidrug-resistant”

Line 60; “According with” should be “According to”

Line 63; “Escherichia coli” abbreviation (E.coli)

Line 65; “diarrhea” should be “diarrhoea”

Line 69; “Extended-spectrum β -lactamase”

Line 76; remove “there are some recent available data in Africa suggest”

Line 129; abbreviation E.coli 

The Authors should carefully check the text and do the English revision.

Author Response

The paper title “First European Report of Escherichia coli producing  Extended-Spectrum β-lactamases (ESBL) from camels in Canary Islands: Animal-Human interface in antibiotic resistance” is an interesting article that provides new and important insights about ESBL spread isolated from camels on the Canary Islands, but the article needs a big, careful revision from the Authors. A lot of valuable work was done but the article has a lot of misunderstandings and mistakes. General comments are that the Authors should read the Instructions for Authors in “Animals” and prepare an article according to them eg. e-mail after affiliation, in “Abstract”, is a reference in line 41, the References are wrong cited in the text: should be [1-4], line 57.

Answer: Thank you for the important comments. We try to follow the Animals template but there are different authors with different affiliations. So, different emails in the same affiliations. We added the emails in the manuscript.

“Material and Methods”

“Animals and sampling” – “One faecal sample per animal was obtained rectally” what with Ethical proof ?

Answer:  Every year the Canary Islands’ animal health authorities are obliged to carry out a livestock operation to combat tuberculosis among other diseases. This EU official directive is applied also to dromedaries in the Canary Islands. Then, all the animals must pass through the handling system at the moment of reading of the intradermal tuberculin test completed by an official veterinarian. One member of our team (Dr. Corbera, who is also a Vet) carried out the sampling at the same moment of the official operation, in a random manner in the two farms included in this study. Both owners signed an agreement with the ULPGC and an authorization for the sampling in theirs animals.

“Escherichia coli isolation” I understand that the Authors put faecal samples from swabs on MacConkey ?  The faecal samples were I need a few words of explanation of this procedure.

Answer: At the laboratory, the faecal samples were previously diluted and after they were spread onto MC agar plates, then either supplemented or not supplemented with 2 μg/ml cefotaxime, in order to recover susceptible and resistant E. coli isolates.

“Antibiotic resistance genes”  as well as the sxt1,2 genes were also analyzed” this “sxt1,2” I once again need a few words of explanation

Answer: We tested different genes, including shiga toxin associated with stx1 and stx2 in all samples. There is a mistake in the text: not “sxt” but “stx”. The change was done.

All article needs English correction. For detailed comments, see minor comments

Minor comments

Line 4; Canary Islands” the authors should check “the Canary Islands”

Answer: The change was done in the title.

Line 26; abbreviation Escherichia coli (E. coli)

Answer: The change was done in the manuscript.

Line 27; the spelling of “harbor” is a non-British variant, the authors should check if in all text should be “harbour”

Answer: The change was done in the manuscript in lines 29 and 200.

Line 27; “multidrug resistant” is missing a hyphen

Answer: The change was done in the manuscript.

Line 29; “Canary Islands” the authors should check “the Canary Islands”

The change was done in the manuscript.

Line 36; “Objective: The aim of this work was” “This work aimed”

Answer: The change was done in the manuscript.

Line 37; “Canary Islands” the authors should check “the Canary Islands”

Answer: The change was done in the manuscript.

Line 38; the authors should remove “apparently” it is not necessary in this place

Answer: The change was done in the manuscript.

Line 41; The authors should check “the disk diffusion”

Answer: The change was done in the manuscript.

Line 43; What the authors meant by “sxt1,2”

Answer: We would like to say stx gene (shiga toxin). The change was done in the manuscript.

Line 45; “Escherichia coli” The authors should decide abbreviation or not Line 36-37

Answer: The change was done in the manuscript (abbreviation).

Line 52; proposition “associated to” should be “associated with”

Answer: The change was done in the manuscript.

Line 59; “multidrug resistant” is missing a hyphen “multidrug-resistant”

Answer: The change was done in the manuscript.

Line 60; “According with” should be “According to”

Answer: The change was done in the manuscript.

Line 63; “Escherichia coli” abbreviation (E.coli)

Answer: The change was done in the manuscript.

Line 65; “diarrhea” should be “diarrhoea”

Answer: The change was done in the manuscript.

Line 69; “Extended-spectrum β -lactamase”

Answer: The change was done in the manuscript.

Line 76; remove “there are some recent available data in Africa suggest”

Answer: The change was done in the manuscript.

Line 129; abbreviation E. coli

Answer: The change was done in the manuscript.

Round 2

Reviewer 5 Report

Literature should be done according to instruction